# VISAGNN: Versatile Staleness-Aware Training for Efficient Large-Scale GNNs

## Abstract

Graph Neural Networks (GNNs) have shown exceptional success in graph representation learning and a wide range of real-world applications. However, scaling deeper GNNs poses challenges due to the neighbor explosion problem when training on large-scale graphs. To mitigate this, a promising class of GNN training algorithms utilizes historical embeddings to reduce computation and memory costs while preserving the expressiveness of the model. These methods leverage historical embeddings for out-of-batch nodes, effectively approximating full-batch training without losing any neighbor information—a limitation found in traditional sampling methods. However, the staleness of these historical embeddings often introduces significant bias, acting as a bottleneck that can adversely affect model performance. In this paper, we propose a novel VersatIle Staleness-Aware GNN, named VISAGNN, which dynamically and adaptively incorporates staleness criteria into the large-scale GNN training process. By embedding staleness into the message-passing mechanism, loss function, and historical embeddings during training, our approach enables the model to adaptively mitigate the negative effects of stale embeddings, thereby reducing estimation errors and enhancing downstream accuracy. Comprehensive experiments demonstrate the effectiveness of our method in overcoming the limitations of existing historical embedding techniques, highlighting its superior performance and efficiency on large-scale benchmarks, as well as significantly accelerated convergence. We will make the code publicly available upon acceptance of the work.

## 1 Introduction

Graph Neural Networks (GNNs) have proven to be highly effective tools for learning representations from graph-structured data (Hamilton, 2020; Ma & Tang, 2021), excelling in tasks such as node classification, link prediction, and graph classification (Kipf & Welling, 2016; Gasteiger et al., 2019; Veličković et al., 2017; Wu et al., 2019). They have also been successfully applied in real-world scenarios like recommendation systems, biological molecule modeling, and transportation networks (Tang et al., 2020; Sankar et al., 2021; Fout et al., 2017; Wu et al., 2022). However, the scalability of GNNs is challenged by their recursive message-passing process, which results in the neighborhood explosion problem. This issue arises because the number of neighbors involved in mini-batch computations grows exponentially with the number of GNN layers (Hamilton et al., 2017; Chen et al., 2018; Han et al., 2023), making it difficult for deeper GNNs to capture long-range dependencies on large graphs. Such long-range information is known to enhance GNN performance (Gasteiger et al., 2018; Gu et al., 2020; Liu et al., 2020; Chen et al., 2020a; Li et al., 2021; Ma et al., 2020; Pan et al., 2020; Zhu et al., 2021; Chen et al., 2020b), but the neighborhood explosion problem limits the ability of GNNs to handle large-scale graphs within the constraints of GPU memory and computational resources during training and inference. This bottleneck significantly hampers the expressive power of GNNs and their applicability to large-scale graphs.

Various approaches have been developed to enhance the scalability of GNNs, including sampling techniques (Hamilton et al., 2017; Chen et al., 2018; Chiang et al., 2019; Zeng et al., 2020), pre- and post-computing strategies (Wu et al., 2019; Rossi et al., 2020; Sun et al., 2021; Huang et al., 2020), and distributed learning (Chai et al., 2022; Shao et al., 2022). Among these, sampling methods are widely used to address the neighborhood explosion problem in large-scale GNNs due to their simplicity and promising results. However, sampling methods often discard information from

unsampled neighbors during training, and because nodes in a graph are interconnected and cannot simply be treated as independent and identically distributed (*iid*), this leads to estimation variance in embedding approximation and an inevitable loss of accurate graph information.

To address this issue, historical embedding methods have been proposed, such as VR-GCN (Chen et al., 2017), MVS-GCN (Cong et al., 2020), GAS (Fey et al., 2021), GraphFM (Yu et al., 2022) and Refresh (Huang et al., 2023). These methods use historical embeddings of unsampled neighbors as approximations of their true aggregated embeddings. During each training iteration, they store intermediate node embeddings at each GNN layer as historical embeddings, which are then utilized in subsequent iterations. This approach effectively mitigates the neighbor explosion problem and reduces the variance associated with sampling methods by preserving all neighbor information. The historical embeddings can be stored offline on CPU memory or disk, conserving GPU memory. These approaches avoid ignoring any nodes or edges, thereby reducing variance and maintaining the expressiveness of the backbone GNNs while achieving strong scalability and efficiency.

While using historical embeddings can provide several benefits, their quality is a crucial determinant of overall performance. Specifically, the discrepancy between a true node embedding and its corresponding historical embedding, which we refer to as the staleness of the historical embeddings, becomes a critical factor since the historical embedding serves as an approximation of the true one. This phenomenon is particularly evident in large-scale datasets, as the update speed of historical embeddings lags far behind that of model parameters. As a result, these historical embeddings become highly stale and exhibit significant discrepancies from the true embeddings. Consequently, historical embedding methods often suffer from substantial degradation in both prediction accuracy and convergence speed when compared to vanilla sampling methods like GraphSAGE (Hamilton et al., 2017), which do not rely on historical embeddings. Thus, staleness becomes the primary bottleneck for these methods.

Motivated by our findings and analysis, effectively utilizing staleness to leverage fresh embeddings while minimizing the impact of stale embeddings has become a critical issue. Therefore, we propose a Versatile Staleness-Aware GNN (VISAGNN), which incorporates three key components: (1) *Dynamic Staleness Attention*: We introduce a novel staleness-based weighted message-passing mechanism that uses staleness scores as a metric to dynamically determine the importance of each node during message passing; (2) *Staleness-aware Loss*: We design a regularization term based on staleness criterion to be included in the loss function, explicitly reducing the influence of staleness on the model; (3) *Staleness-Augmented Embeddings*: We offer a straightforward solution by directly injecting staleness into the node embeddings. Our proposed framework is highly flexible, orthogonal, and compatible with various sampling methods and historical embedding techniques. Comprehensive experiments demonstrate that it further enhances existing historical embedding methods, accelerating convergence while maintaining strong efficiency.

## 2 RELATED WORK

In this section, we summarize related works on the scalability of large-scale GNNs with a focus on sampling methods.

**Sampling methods.** Sampling methods utilize mini-batch training strategies by selecting a subgraph as a small batch, reducing computation and memory requirements. These methods fall into three main categories: node-wise sampling, layer-wise sampling, and subgraph-wise sampling.

(1)*Node-wise sampling*: This approach samples a fixed number of neighbors per hop, as seen in models like GraphSAGE (Hamilton et al., 2017), PinSAGE (Ying et al., 2018), and GraphFM-IB (Yu et al., 2022). However, because it involves dropping unsampled nodes and edges, it introduces bias and variance. Additionally, while it helps mitigate the neighbor explosion problem, it doesn't entirely solve it since the number of neighbors still grows exponentially.

(2)*Layer-wise sampling*: It addresses the neighbor explosion problem by fixing the number of sampled neighbors per layer. For example, FastGCN (Chen et al., 2018) treats message passing from an integral perspective, independently sampling nodes in each GNN layer using importance sampling. LADIES (Zou et al., 2019) incorporates inter-layer correlations by restricting the sampled nodes to those in the union of the neighbors of already sampled nodes. ASGCN (Huang et al., 2018) designs the sampling probability of lower layers based on the upper layers. However, the adjacency matrix

generated by layer-wise sampling tends to be sparser than that of other methods, often leading to suboptimal performance.

(3)*Subgraph sampling*: It involves directly sampling subgraphs from the entire graph as mini-batches and then performing message passing on these subgraphs. This method effectively addresses the neighbor explosion problem, as the GNN operates only on the sampled subgraph during computation. ClusterGCN (Chiang et al., 2019) pioneered this approach by clustering the graph into subgraphs, with each mini-batch constructed from several clusters. GraphSaint (Zeng et al., 2020) extended this by incorporating various samplers to construct subgraphs, reducing bias and using importance sampling to minimize variance. However, these methods can still suffer from high variance due to the ignored edges between subgraphs.

**Historical embedding methods.** While sampling methods effectively alleviate the neighbor explosion problem, they often suffer from performance degradation due to the variance introduced by dropping nodes and edges. To address this issue, some approaches have started using historical embeddings as an approximation for the true embeddings obtained from full-batch computation. This allows them to avoid dropping any nodes or edges while still reducing memory costs by limiting the number of sampled neighbors. VR-GCN(Chen et al., 2017) was the first to propose using historical embeddings for out-of-batch nodes to reduce variance while limiting the number of sampled neighbors per hop to reduce memory consumption. MVS-GCN(Cong et al., 2020) improved this approach with a one-shot sampling strategy, eliminating the need for nodes to recursively explore their neighborhoods in each layer. GNNAutoScale (Fey et al., 2021) restricts the receptive field to direct one-hop neighbors, enabling constant GPU memory consumption while still preserving all relevant neighbor information. GraphFM-OB(Yu et al., 2022) enhanced performance by incorporating feature momentum. LMC (Shi et al., 2023) considered backward propagation, retrieving discarded embeddings during backward passes, which improved performance and accelerated convergence.

Although these historical embedding approaches are promising due to their strong performance and scalability, they are limited by approximation errors caused by the staleness of the historical embeddings. This issue becomes more pronounced with large-scale datasets. To address this, GAS (Fey et al., 2021) utilizes graph clustering to reduce inter-connectivity, a proven factor contributing to staleness, and applies regularization to limit significant changes in model parameters, thereby mitigating approximation errors. GraphFM-OB(Yu et al., 2022) compensates for staleness by leveraging feature momentum for in-batch nodes nd out-of-batch nodes. Despite these efforts, these methods only tackle the issue superficially, resulting in minimal performance improvements. Refresh (Huang et al., 2023) introduces a staleness score, which quantifies the degree of staleness, and avoids using stale embeddings to alleviate this issue. However, it may result in the loss of some direct neighbor information, introducing significant bias.

**Other scalable designs** Pre-computing or post-computing methods aim to offload the computationally intensive feature aggregation to the CPU. This can be achieved by pre-computing the message passing before training (Wu et al., 2019; Rossi et al., 2020; Sun et al., 2021; Zhang et al., 2022; Bojchevski et al., 2020), or through post-processing with label propagation, as demonstrated by methods like (Huang et al., 2020). Despite their benefits, these approaches often lose the advantages of end-to-end training. Additionally, distributed methods enhance scalability by distributing large graphs across multiple GPUs to parallelize GNN training, as demonstrated in (Chiang et al., 2019; Chai et al., 2022; Shao et al., 2022). However, they usually incur significant communication costs between GPUs.

## 3 METHODOLOGY

In this section, we first mathematically formulate historical embedding methods and then theoretically demonstrate that staleness is a key factor in the effectiveness of these methods. Then, we present a novel VersatIle Staleness-Aware GNN (VISAGNN), that dynamically incorporates staleness into the training process from multiple sensory perspectives, utilizing the staleness criterion as a metric to prioritize fresher embeddings over stale ones.

### 3.1 STALENESS OF HISTORICAL EMBEDDING METHODS

Sampling is used to generate mini-batches for message passing to address the scalability challenge in large-scale graphs:

$$h_i^{(l+1)} = g_\theta^{(l+1)}(h_i^l, [h_j^l]_{j \in \mathcal{N}(i)}) \approx g_\theta^{(l+1)}(h_i^l, [h_j^l]_{j \in \mathcal{N}(i) \cap B}) \tag{1}$$

Here, $h_i^l$ represent the feature embedding of the in-batch node $i$ at the $l$-th layer, and $g_\theta^{(l+1)}$ denote the message-passing update function at the $l+1$-th layer with parameters $\theta$. The set $\mathcal{N}(i) \cap B$ refers to the in-batch 1-hop neighborhood of node $i$. However, the large variance arises because the out-of-batch neighbors $[h_j^l]_{j \in \mathcal{N}(i) \setminus B}$ are not considered during aggregation.

To address this issue, historical embedding methods utilize historical embeddings $[\bar{h}_j^l]_{j \in \mathcal{N}(i) \setminus B}$ to approximate the embeddings of out-of-batch nodes $[h_j^l]_{j \in \mathcal{N}(i) \setminus B}$ at each layer, providing an approximation of full-batch aggregation. The feature memory is then updated for future use, using only the in-batch node embeddings $\bar{h}_i^{l+1} = h_i^{l+1}$. The message-passing process can be expressed as:

$$h_i^{(l+1)} = g_\theta^{(l+1)}(h_i^l, [h_j^l]_{j \in \mathcal{N}(i)}) \tag{2}$$

$$= g_\theta^{(l+1)}(h_i^l, \underbrace{[h_j^l]_{j \in \mathcal{N}(i) \cap B}}_{\text{in-batch neighbors}} \cup \underbrace{[h_j^l]_{j \in \mathcal{N}(i) \setminus B}}_{\text{out-of-batch neighbors}}) \tag{3}$$

$$\approx g_\theta^{(l+1)}(h_i^l, \underbrace{[h_j^l]_{j \in \mathcal{N}(i) \cap B}}_{\text{in-batch neighbors}} \cup \underbrace{[\bar{h}_j^l]_{j \in \mathcal{N}(i) \setminus B}}_{\text{historical embeddings}}), \tag{4}$$

While using historical embeddings as approximations helps retain information for out-of-batch nodes and ensures constant memory usage (Fey et al., 2021), large approximation errors in certain stale embeddings can significantly degrade model performance. To highlight this issue and motivate our approach, we first present a theoretical analysis showing that the approximation error of the final embeddings is upper bounded by the staleness. Our analysis adheres to the assumptions outlined in previous work (Fey et al., 2021).

**Theorem 1** (Embeddings Approximation Error). *Assuming a L-layers GNN $g_\theta^{(l)}(h)$ with a Lipschitz constant $\beta^{(l)}$ for each layer $l = 1, \ldots, L$, and $\mathcal{N}(i)$ is the set of neighbor nodes of $i$, $\forall i \in V$. $\|\bar{h}^{(l)} - h^{(l)}\|$ represents the distance between the historical embeddings and the true embeddings, which corresponds to the staleness. The approximation error of the final layer embeddings $\tilde{h}_i^{(L)}$ is then upper bounded by:*

$$||\tilde{h}_i^{(L)} - h_i^{(L)}|| \leq \sum_{k=1}^{L} (\prod_{l=k+1}^{L} \beta^{(l)} |\mathcal{N}(i)| * ||\tilde{A}_{i,}|| * ||\bar{h}^{(k-1)} - h^{(k-1)}||).$$

The proof of the above theorem can be found in Appendix A. From the above theorem, we can observe that the distance between the final layer's embeddings produced by historical embedding methods and full aggregations is bounded by a cumulative sum of the per-layer approximation error $||\bar{h}^{(k-1)} - h^{(k-1)}||$. To prevent the accumulation of staleness across layers from having a significantly negative impact on the quality of the final embeddings, reducing the impact of staleness at each layer becomes a crucial issue.

From Theorem 1, $||\bar{h}^{(k-1)} - h^{(k-1)}||$ directly measures the staleness which is the distance between historical embeddings and true embeddings. However, it is impractical to recompute the true embedding $h^{(k-1)}$ at every iteration due to the significantly higher computational overhead involved. Hence, we adopt a lightweight approach from existing works (Huang et al., 2023) by using two indicators to represent the staleness criterion $s_i$: the persistence time $T_i$ and the gradient norm criterion $||\nabla L_\theta(h_i)||$. We cache these two indicators from each layer along with the corresponding historical embeddings for use in our training framework.

The persistence $T_i$ for a specific node $i$ measures how many training iterations the historical embedding remains unchanged before being updated again. Since the historical embedding of a specific node is updated only once per epoch when it serves as a target node and remains the same in the cache

Figure 1: Three key designs in VISAGNN. (1) **Augmented Embeddings**: VISAGNN offers two ways to integrate staleness criterion into historical embeddings. (2) **Dynamic Staleness Attention**: VISAGNN performs weighted message passing based on both feature embeddings and staleness criterion. (3) **Staleness-aware loss**: A regularization term based on staleness is incorporated into the loss function in VISAGNN.

for the rest of the iterations, while the model parameters continue to update throughout all training iterations, $T_i$ reflects the gap between the update frequencies of the historical embeddings and the model parameters, directly capturing staleness in a straightforward manner. A high persistence value indicates that the historical embedding has not been updated recently, leading to stronger feature staleness.

In addition, the norm of the gradient metric $||\nabla L_\theta(h_i)||$ reflects the extent of changes in the model parameters, which can also indicate the staleness. A small gradient magnitude suggests that the model parameters are not changing significantly, leading to stable node embeddings throughout the training iterations. Consequently, the estimation error of the historical embeddings is likely to be small, leading to minimal staleness.

## 3.2 DYNAMIC STALENESS ATTENTION

As introduced in Section 1, staleness becomes a bottleneck for existing historical embedding methods. While existing works like Refresh (Huang et al., 2023) utilize staleness criteria as thresholds to evict embeddings that have not been recently updated or are unstable, simply discarding these embeddings based on staleness can introduce significant bias. This approach also makes the model overly sensitive to the fixed staleness threshold, as the embeddings of any nodes with staleness exceeding the threshold are discarded, even though their degrees of staleness may vary. Consequently, this motivates us to investigate the dynamic integration of staleness into the training process, allowing us to consider staleness while retaining essential graph features.

We propose a staleness-aware attention mechanism by incorporating the staleness criterion into the traditional attention formulation. This mechanism adjusts attention coefficients based on both the node's current features and staleness. The attention coefficients for in-batch neighbors $\alpha_{ij}^{\text{in}}$ and out-of-batch neighbors $\alpha_{ij}^{\text{out}}$ between node $i$ and node $j$ at epoch $t$ are formulated as follows. We omit the layer number $L$ in $\alpha$ for simplicity:

$$\boldsymbol{\alpha}_{ij}^{\text{out}}(t) = \frac{\exp\left(\text{LeakyReLU}\left(\mathbf{a}^T\left[\mathbf{W}h_i \parallel \mathbf{W}\bar{h}_j\right]\right) - \boldsymbol{\gamma}(t) \cdot s_j \cdot \sigma(c_j - c_{\text{avg}})\right)}{\sum_{k \in \mathcal{N}(i)\setminus B} \exp\left(\text{LeakyReLU}\left(\mathbf{a}^T\left[\mathbf{W}h_i \parallel \mathbf{W}\bar{h}_k\right]\right) - \boldsymbol{\gamma}(t) \cdot s_k \cdot \sigma(c_k - c_{\text{avg}})\right)} \quad (5)$$

$$\boldsymbol{\alpha}_{ij}^{\text{in}}(t) = \boldsymbol{\alpha}_{ij}^{\text{out}}(t)\big|_{s_j, s_k = 0, \bar{h} = \tilde{h}} \quad (6)$$

$$\tilde{h}_i^{(L)} = \phi\left(\sum_{j \in \mathcal{N}(i) \cap B} \boldsymbol{\alpha}_{ij}^{L-1,\text{in}}\mathbf{W}h_j^{(L-1)}, \sum_{j \in \mathcal{N}(i)\setminus B} \boldsymbol{\alpha}_{ij}^{L-1,\text{out}}\mathbf{W}\bar{h}_j^{(L-1)}\right) \quad (7)$$

where $\mathbf{W}$ is the weight matrix applied for each nodes for feature transformation, $\mathbf{a}$ is a learnable weight vector used to compute the attention score between two nodes, similar as GAT. The operator $\|$ denotes concatenation. The term $s_j$ represents the staleness criterion of node $j$, reflecting how outdated the embedding of node $j$ is. The function $\sigma$ represents the nonlinear function, we specifically utilize sigmoid function in this paper, defined as $f(x) = \frac{1}{1+e^{-x}}$. The value $c_j$ is a centrality measure for node $j$, while $c_{avg}$ denotes the average centrality measure across the graph. In this paper, we use node degree as centrality metric to evaluate the importance of each node. The time-dependent coefficient $\gamma(t) = \frac{\beta}{t}$, where $t$ is the current epoch, $\beta$ is a learnable scaling factor that controls how quickly $\gamma(t)$ decreases with the training process for each node. It modulates the impact of the staleness on attention during the training process. $\phi$ is a non-linear activation function. Note that $\alpha_{ij}^{\text{in}}(t)$ degenerates into the traditional attention scores in GAT when staleness equals 0, which aligns with our intuition.

The core of our design revolves around the term $-\gamma(t) * s_j * \sigma(c_j - c_{avg})$, which consists of three components:

**(1) Staleness Criterion:** $s_j$ represents the staleness of each node embedding, which is the key for achieving staleness-aware attention. In our implementation, we choose to use gradient criterion $||\nabla L_\theta(h_i)||$. The gradients at any layer are obtained from backward propagation when the corresponding node was included into the computation graph previously.

**(2) Centrality:** After considering the impact of feature embeddings, centrality $c$ is introduced to incorporate the graph structure. The motivation is that if a stale node is important, the negative effects caused by staleness will be amplified. Specifically, when the degree of a node is high and the staleness is also high, this term significantly penalizes and reduces the attention coefficient to mitigate the impact of staleness, as these stale embeddings are propagated through many neighboring nodes. Conversely, if the staleness is low, the node's embedding is fresh and will not cause significant negative effects, allowing it to be effectively utilized. When the degree is low, these nodes are less critical to the final representation, so staleness may have a smaller impact. Furthermore, we choose to use relative centrality by subtracting the average node degree of the graph from each node's degree, $c_j - c_{\text{avg}}$, to prevent the issue that graphs with dense connections naturally have high node degrees. We then use the sigmoid function to further reduce the scale impact.

**(3) Decay Coefficient:** We also introduce a function $\gamma(t)$ related to the training process as a coefficient for the staleness term. The reason for this is that as training progresses, the model parameters gradually converge, leading to minimal updates of the embeddings in the final few epochs. Therefore, the influence of staleness should not play a significant role when calculating the attention score. Although there are many feasible designs, we directly used $\frac{\beta}{t}$ for the sake of simplicity.

## 3.3 STALENESS-AWARE LOSS

In addition to the proposed dynamic staleness attention, we also incorporate staleness into the optimization process as a regularization term to more effectively mitigate its effects. However, the gradient criterion $||\nabla L_\theta(h_i)||$ for staleness is not feasible to use since the loss has not yet been computed. From Theorem 1, we find that the final representation contains the accumulated staleness from all layers, allowing it to effectively represent staleness. Hence, we choose to utilize the feature embeddings of in-batch nodes at last layer between two consecutive epochs for our design. The staleness-aware regularization term is defined as follows:

$$\mathcal{L}_{\text{stale}} = \sum_{i \in B} ||h_{i,k}^{(L)} - h_{i,k-1}^{(L)}||^2 \tag{8}$$

where $h_{i,k}^{(L)}$ represents the feature embedding of node $i$ at the final layer $L$ during epoch $k$. This design is based on the observation that as training progresses, model parameters tend to converge, resulting in smaller gradient values and fewer updates to the embeddings in later epochs. Consequently, the difference between final representations from consecutive epochs becomes progressively smaller, particularly after the model has been trained for several epochs. This aligns with our earlier conclusion that the influence of staleness diminishes as training progresses. Another advantage of this design is that it does not introduce any additional computational overhead.

By jointly optimizing both the downstream tasks and the staleness issue, the gradient also becomes staleness-aware, which better mitigates the negative effects of staleness on the model's performance. For the sake of simplicity, we define the overall training loss as follows:

$$\mathcal{L} = \mathcal{L}_{\text{task}} + \lambda \cdot \mathcal{L}_{\text{stale}} \tag{9}$$

where $\lambda$ is a hyperparameter that controls the trade-off between the task-specific loss and the penalty for staleness. $\mathcal{L}_{\text{task}}$ is the task-specific loss, such as cross-entropy in node classification.

### 3.4 STALENESS-AUGMENTED EMBEDDINGS

We further enhance the model's performance by incorporating staleness awareness through the direct injection of staleness criterion into node embeddings. We present two implementation methods: *concatenation* and *summation*.

**Concatenation**: We treat staleness as an additional dimension of feature and concatenate it with the historical embeddings at each layer. To ensure that the impact of staleness is appropriately balanced, we first normalize the staleness criterion using a log normalization technique. This approach helps to mitigate the influence of imbalanced distributions, such as extremely high staleness criterion values, ensuring that stale embeddings do not dominate the feature representation. It also prevents staleness from being overly influential due to differences in scale when combined with the node features. The augmented embeddings can be represented as:

$$\bar{h_j}' = \text{Concat}\left(\bar{h_j}, log(1 + s_j)\right) \tag{10}$$

**Summation**: This approach differs from simple concatenation. We combine the staleness criterion with the node features through a non-linear transformation, allowing the model to learn a weighted combination of the node's inherent features and its staleness, potentially capturing their interactions and enhancing the expressiveness of the learned node representations. Specifically, suppose $\mathbf{W}_s$ is a learnable weight matrix, the transformation can be represented as:

$$\bar{h_j}' = \bar{h_j} + \phi(\mathbf{W}_s \cdot s_j) \tag{11}$$

Similarly, we use $||\nabla L_\theta(h_i)||$ as the staleness criterion $s_i$. This choice is based on our experiments, which indicate that extreme values in the persistence time $T_i$ can adversely affect aggregation, causing the model to overly focus on the staleness term. Consequently, this negatively affects convergence, especially on very large datasets. However, we still utilize $T_i$ as used in Refresh: We set a dataset-dependent high value threshold $G_{\text{thres}}$, a small portion of nodes whose persistence times are significantly larger during each training iteration to further mitigate the impact of staleness while most nodes are not affected by this criterion.

## 4 EXPERIMENTS

In this section, we present experiments that demonstrate the effectiveness of our proposed algorithms in enhancing performance, improving efficiency, and accelerating convergence.

### 4.1 PERFORMANCE

**Experimental setting.** We present a performance comparison against major baselines, including several classical GNN models such as GCN (Kipf & Welling, 2016), GraphSAGE (Hamilton et al., 2017), FastGCN (Chen et al., 2018), LADIES (Zou et al., 2019), Cluster-GCN (Chiang et al., 2019), GraphSAINT (Zeng et al., 2020), and SGC (Wu et al., 2019). Additionally, we include state-of-the-art methods for historical embeddings such as VR-GCN (Chen et al., 2017), MVS-GCN (Cong et al., 2020), GNNAutoScale (GAS) (Fey et al., 2021), GraphFM (Yu et al., 2022), Refresh (Huang et al., 2023) and LMC (Shi et al., 2023). For the last four models, we employ GAT as the GNN backbone to ensure a fair comparison with our proposed methods. We conduct experiments on three widely-used large-scale graph datasets: REDDIT, ogbn-arxiv, and ogbn-products (Hu et al., 2020). We denote the augmentation strategies of concatenation and summation introduced in Section 3 as VISAGNN-Cat and VISAGNN-Sum, respectively. The performance results are reported in Table 1, where OOM

stands for out-of-memory. Additionally, we provide a direct and clear performance comparison with other historical embedding methods on a significantly larger dataset, ogbn-papers100M. The results are shown in Table 2. For all baselines, we follow the configurations provided in their respective papers and official repositories.

VISAGNN's hyperparameters are tuned from the following search space: (1) learning rate: $\{0.01, 0.001, 0.0001\}$; (2) weight decay: $\{0, 5e-4, 5e-5\}$; (3) dropout: $\{0.1, 0.3, 0.5, 0.7\}$; (4) propagation layers : $L \in \{1, 2, 3\}$; (5) MLP hidden units: $\{256, 512\}$; (6) $\lambda \in \{0.1, 0.3, 0.5, 0.8\}$.

**Performance analysis.** From the results of the performance comparison, we can draw the following observations:

• None of the existing historical embedding methods consistently outperform classical models on large-scale datasets such as ogbn-products, and they only surpass other scalable methods by a small margin on other datasets. This is due to the slower update of historical embeddings compared to model parameters, especially given the large number of batches in a single training epoch, highlighting staleness as a significant bottleneck for all historical embedding techniques. It is worth noting that while Refresh performs well on large-scale datasets, it falls significantly behind other baselines when staleness is not dominant (ogbn-arxiv). This is because it simply evicts some important neighbors, which can potentially introduce significant bias, reinforcing our claim made in Section 2.

Table 1: Accuracy comparison (%) with major baselines.

| Method | GNNs | # nodes 230K # edges 11.6M REDDIT | 169K 1.2M ogbn arxiv | 2.4M 61.9M ogbn products |
|---|---|---|---|---|
| Scalable | GraphSAGE | 95.4 | 71.5 | 78.7 |
| | FastGCN | 93.7 | — | — |
| | LADIES | 92.8 | — | — |
| | Cluster-GCN | 96.6 | — | 79.0 |
| | GraphSAINT | **97.0** | — | 79.1 |
| | SGC | 96.4 | — | — |
| | VR-GCN | 94.1 | 71.5 | 76.3 |
| | MVS-GNN | 94.9 | 71.6 | 76.9 |
| Full Batch | GCN | 95.4 | 71.6 | OOM |
| | GAT | 95.7 | 71.5 | OOM |
| | APPNP | 96.1 | 71.8 | OOM |
| Historical | GAS | 95.7 | 71.7 | 77.0 |
| | GraphFM | 95.6 | 71.9 | 77.2 |
| | Refresh | 95.4 | 70.4 | 78.7 |
| | LMC | 96.2 | 72.2 | 77.5 |
| Ours | **VISAGNN-Cat** | 96.5 | 73.0 | 79.9 |
| | **VISAGNN-Sum** | 96.6 | **73.2** | **80.2** |

• When comparing performance on large-scale datasets, the proposed VISAGNN outperforms all baselines on ogbn-arxiv, ogbn-products and ogbn-papers100M, particularly in comparison to state-of-the-art historical embedding methods, while achieving comparable results on Reddit. Notably, VISAGNN shows substantial improvements on large scale datasets, highlighting the necessity and significance of the staleness-aware techniques we introduced, especially under conditions of increased staleness. Furthermore, VISAGNN-Sum surpasses VISAGNN-Cat, indicating that using a learnable fully connected layer is more effective for integrating staleness information into node embeddings, resulting in improved final representations.

• The strategies we proposed in VISAGNN can be integrated with various baselines. For instance, in LMC, historical gradients also encounter the issue of staleness, which dynamic attention can help alleviate during gradient message passing. This advantage underscores the flexibility and adaptability of our model.

## 4.2 EFFICIENCY ANALYSIS

In this section, we provide an efficiency analysis, including memory usage and total running time on the ogbn-arxiv and ogbn-products datasets, comparing our method against one classical scalable GNN, GraphSAGE, and two historical

Table 2: Prediction accuracy (%) comparison with other baselines on ogbn-papers100M

| Method | GAS | FM | Refresh | LMC | VISAGNN |
|---|---|---|---|---|---|
| Acc(%) | 57.5 | 58.6 | 65.4 | 61.3 | **67.5** |

embedding methods, GAS and Refresh, as shown in Table 3. Note that we exclude system-level optimizations from Refresh to ensure a fair comparison. All experiments were conducted on a single GPU. To ensure a fair comparison, we employed the official implementations for all baseline methods and kept the hyperparameters consistent. For GAS and Refresh, we used GAT as the GNN backbone since both methods also leverage attention mechanisms.

From the results, we observe that GraphSAGE still suffers from the neighbor explosion problem, leading to out-of-memory (OOM) errors on ogbn-products and significantly higher memory costs for ogbn-arxiv in our experiments. Refresh requires less running time on ogbn-products as it converges more quickly due to the eviction of stale embeddings. However, it takes longer to converge on ogbn-arxiv compared to other models. In contrast, VISAGNN maintains nearly the same memory usage as GAS and Refresh while accelerating the training process. This improvement is attributed to VISAGNN's ability to achieve the fastest convergence among all historical embedding baselines, requiring substantially fewer epochs to reach convergence. Moreover, while VISAGNN-Sum incurs slightly higher memory costs and running time than VISAGNN-Cat due to the inclusion of a fully connected layer, it demonstrates improved performance.

Table 3: Memory usage (MB) and running time (seconds) on ogbn-arxiv and ogbn-products.

| Dataset | MEMORY (MB) | | | | | TIME (S) | | | | |
|---|---|---|---|---|---|---|---|---|---|---|
| | Sage | GAS | Refresh | VISAGNN-Cat | VISAGNN-Sum | SAGE | GAS | Refresh | VISAGNN-Cat | VISAGNN-Sum |
| ogbn-arxiv | 2997 | 767 | 791 | 813 | 869 | 21 | 40 | 49 | 22 | 26 |
| ogbn-products | OOM | 8886 | 8933 | 8982 | 9017 | N/A | 2522 | 2178 | 1303 | 1380 |

## 4.3 CONVERGENCE ANALYSIS

We provide a convergence analysis by comparing the test accuracy over time for baselines, including GAS, Refresh, and our proposed VISAGNN, on the ogbn-arxiv and ogbn-products datasets. The results in Figure 2 and 3 (S stands for summation, C stands for concatenation) reveal that when staleness is not significant (as in the ogbn-arxiv case), Refresh performs poorly because it loses information from neighbors. However, when staleness is significant (as in the ogbn-products case), GAS's convergence is heavily affected by staleness. In contrast, our model achieves faster convergence and superior performance on both cases by effectively accounting for varying levels of staleness in the historical embeddings during training, as introduced in Section 3. This advantage becomes especially clear on large datasets, where staleness tends to be more severe. Overall, these findings show that our algorithm not only improves performance but also accelerates convergence.

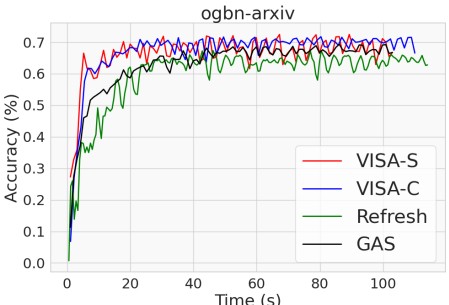

Figure 2: Test Accuracy on ogbn-arxiv     Figure 3: Test Accuracy on ogbn-products

## 4.4 ABLATION STUDY

### 4.4.1 HYPERPARAMETERS

Given the three novel techniques introduced in our VISAGNN, we conducted an ablation study on the ogbn-arxiv and ogbn-products datasets to identify which technique contributes the most to the final performance. For simplicity, we denote the dynamic attention, staleness-aware loss, and augmented embeddings as "att," "loss," and "emb," respectively. We use summation here for the augmented embedding. To ensure a fair comparison, all other hyperparameters are kept consistent, and we test various combinations of the three proposed strategies.

From Table 4, we observe that the best performance occurs when all three techniques are applied. Specifically, the dynamic attention mechanism contributes the most, as it explicitly considers the

staleness of each historical embedding during message passing and integrates this information into the training process, preventing overly stale embeddings from harming the final representation. Additionally, the proposed loss term enhances model performance by accounting for staleness in each training iteration, injecting this information into the gradient through backpropagation, thereby promoting staleness awareness in the model.

### 4.4.2 STALENESS RESISTANCE

Previous results demonstrated that our model effectively mitigates the negative impact of stale embeddings. In this section, we further demonstrate our model's effectiveness in mitigating the adverse effects of staleness by conducting experiments with varying levels of staleness through different batch sizes on ogbn-arxiv and ogbn-products. When the batch size is small, the staleness be-

Table 4: Prediction accuracy (%) comparison of different components

| Method | ogbn-arxiv | ogbn-products |
|---|---|---|
| VISAGNN w/o att | 72.0 | 77.8 |
| VISAGNN w/o loss | 72.6 | 79.2 |
| VISAGNN w/o emb | 72.9 | 79.8 |
| VISAGNN | 73.2 | 80.2 |

comes significant because there are more parameter updates within an epoch, while the historical embeddings are updated only once. We compare our model with existing representative historical embedding methods: GAS, GraphFM, LMC, and Refresh. The results are presented in Table 5. In these experiments, we strictly adhere to the settings outlined in their papers and official repositories. The graphs undergo pre-clustering using METIS (Fey et al., 2021), with the total number of clusters detailed in Table under the label "Clusters." The term "BS" refers to the number of clusters in the current mini-batch.

We observe that the performance of all baselines significantly drops as staleness increases. Specifically, we find that Refresh performs worse when staleness is low, as it directly drops neighbors based on staleness criteria, resulting in the loss of important information from direct neighbors, as discussed in Section 2. In contrast, our algorithms maintain strong performance across all cases, significantly outperforming all baselines, particularly in scenarios with large datasets and small batch sizes where staleness is prominent. For example, we observe a notable performance boost of 2.6% over GAS on the ogbn-products dataset and 3.8% over Refresh on ogbn-arxiv when the batch size is 5. This demonstrates the strong staleness resistance of our model, as all three proposed strategies help make the training process staleness-aware, effectively mitigating the negative impact of stale embeddings on model performance.

Table 5: Accuracy (%) for different batch sizes.

| DATASET | CLUSTERS | BS | GAS | FM | REFRESH | LMC | VISAGNN |
|---|---|---|---|---|---|---|---|
| **Products** | **150** | 5 | $74.5 \pm 0.6$ | $74.8 \pm 0.4$ | $76.1 \pm 0.3$ | $75.0 \pm 0.4$ | $77.1 \pm 0.3$ |
| | | 10 | $75.6 \pm 0.4$ | $76.0 \pm 0.3$ | $77.5 \pm 0.3$ | $76.3 \pm 0.2$ | $79.2 \pm 0.3$ |
| | | 20 | $77.0 \pm 0.3$ | $77.2 \pm 0.2$ | $78.7 \pm 0.2$ | $77.5 \pm 0.3$ | $80.2 \pm 0.2$ |
| **Reddit** | **200** | 20 | $94.8 \pm 0.2$ | $94.7 \pm 0.3$ | $94.9 \pm 0.2$ | $95.0 \pm 0.1$ | $95.7 \pm 0.1$ |
| | | 50 | $95.0 \pm 0.2$ | $95.1 \pm 0.3$ | $95.1 \pm 0.3$ | $95.7 \pm 0.2$ | $96.2 \pm 0.1$ |
| | | 100 | $95.7 \pm 0.1$ | $95.6 \pm 0.2$ | $95.4 \pm 0.2$ | $96.2 \pm 0.1$ | $96.6 \pm 0.2$ |
| **Arxiv** | **40** | 5 | $69.5 \pm 0.4$ | $70.1 \pm 0.3$ | $68.9 \pm 0.2$ | $71.5 \pm 0.2$ | $72.7 \pm 0.2$ |
| | | 10 | $70.1 \pm 0.3$ | $70.5 \pm 0.3$ | $69.2 \pm 0.3$ | $71.8 \pm 0.2$ | $72.9 \pm 0.2$ |
| | | 20 | $71.7 \pm 0.2$ | $71.9 \pm 0.2$ | $70.4 \pm 0.2$ | $72.2 \pm 0.1$ | $73.2 \pm 0.2$ |

## 5 CONCLUSION

Historical embedding methods have emerged as a promising solution for training GNNs on large-scale graphs by solving the neighbor explosion problem while maintaining model effectiveness. However, staleness has become a major limitation of these methods. In this work, we first present a theoretical analysis of this issue and then introduce VISAGNN, a versatile GNN framework that dynamically incorporates staleness criteria into the training process through three key designs. Experimental results show significant improvements over traditional historical embedding methods, particularly in scenarios with pronounced staleness, while accelerating model convergence and preserving good memory efficiency. It provides a flexible and efficient solution for large-scale GNN training.

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

## A    PROOF OF THEOREM

**Notations.** A graph is represented by $\mathcal{G} = (\mathcal{V}, \mathcal{E})$ where $\mathcal{V} = \{v_1, \ldots, v_n\}$ is the set of $n$ nodes and $\mathcal{E} = \{e_1, \ldots, e_m\}$ is the set of $m$ edges. We denote the $d$-dimensional feature vectors of nodes as $\mathbf{X} \in \mathbb{R}^{n \times d}$. The graph structure of $\mathcal{G}$ can be represented by an adjacency matrix $\mathbf{A} \in \mathbb{R}^{n \times n}$, where $\mathbf{A}_{ij} > 0$ when there exists an edge between node $v_i$ and $v_j$, and $\mathbf{A}_{i,j} = 0$ otherwise. The neighboring nodes of node $v$ is denoted by $\mathcal{N}(v)$. The symmetrically normalized graph Laplacian matrix is defined as $L = I - A$ with $\hat{A} = D^{-1/2} A D^{-1/2}$ where $D$ is the degree matrix.

**Theorem 1** (Embeddings Approximation Error). *Assuming a L-layers GNN $g_\theta^{(l)}(h)$ with a Lipschitz constant $\beta^{(l)}$ for each layer $l = 1, \ldots, L$, and $\mathcal{N}^{(l)}$ is the set of neighbor nodes of $i$, $\forall i \in V$. $\|\bar{h}^{(l)} - h^{(l)}\|$ represents the distance between the historical embeddings and the true embeddings, which corresponds to the staleness. The approximation error of the final layer embeddings is then upper bounded by:*

$$\|\tilde{h}_i^{(L)} - h_i^{(L)}\| \le \sum_{k=1}^{L} (\prod_{l=k+1}^{L} \beta^{(l)} |\mathcal{N}(i)| * \|\tilde{\hat{A}}_{i,}\| * \|\bar{h}^{(k-1)} - h^{(k-1)}\|).$$

*Proof.* Suppose $\tilde{g}_\theta^{(l)}$ is a historical embedding-based GNN with L-layers, then the whole GNN model can be defined as $\tilde{h}^{(L)} = \tilde{g}_\theta^{(L)} \circ \tilde{g}_\theta^{(L-1)} \circ \cdots \circ \tilde{g}_\theta^{(1)}$, similarly, the full batch GNN can be defined as: $h^{(L)} = g_\theta^{(L)} \circ g_\theta^{(L-1)} \circ \cdots \circ g_\theta^{(1)}$, then:

$$\|\tilde{h}^{(L)} - h^{(L)}\| = \|\tilde{g}_\theta^{(L)} \circ \tilde{g}_\theta^{(L-1)} \circ \cdots \circ \tilde{g}_\theta^{(1)} - g_\theta^{(L)} \circ g_\theta^{(L-1)} \circ \cdots \circ g_\theta^{(1)}\| \tag{12}$$

$$= \|\tilde{g}_\theta^{(L)} \circ \tilde{g}_\theta^{(L-1)} \circ \cdots \circ \tilde{g}_\theta^{(1)} - \tilde{g}_\theta^{(L)} \circ \tilde{g}_\theta^{(L-1)} \circ \cdots \circ g_\theta^{(1)} \tag{13}$$

$$+ \tilde{g}_\theta^{(L)} \circ \tilde{g}_\theta^{(L-1)} \circ \cdots \circ \tilde{g}_\theta^{(2)} \circ g_\theta^{(1)} - \tilde{g}_\theta^{(L)} \circ \tilde{g}_\theta^{(L-1)} \circ \cdots \circ g_\theta^{(2)} \circ g_\theta^{(1)} - \cdots \tag{14}$$

$$+ \tilde{g}_\theta^{(L)} \circ g_\theta^{(L-1)} \circ \cdots \circ g_\theta^{(1)} - g_\theta^{(L)} \circ g_\theta^{(L-1)} \circ \cdots \circ g_\theta^{(1)}\| \tag{15}$$

$$\le \|\tilde{g}_\theta^{(L)} \circ \tilde{g}_\theta^{(L-1)} \circ \cdots \circ \tilde{g}_\theta^{(1)} - \tilde{g}_\theta^{(L)} \circ \tilde{g}_\theta^{(L-1)} \circ \cdots \circ g_\theta^{(1)}\| + \tag{16}$$

$$\cdots + \|\tilde{g}_\theta^{(L)} \circ g_\theta^{(L-1)} \circ \cdots \circ g_\theta^{(1)} - g_\theta^{(L)} \circ g_\theta^{(L-1)} \circ \cdots \circ g_\theta^{(1)}\| \tag{17}$$

$$= \sum_{k=1}^{L} \left( \prod_{l=k+1}^{L} \beta^{(l)} \|\tilde{g}_\theta^{(k)} \circ g_\theta^{(k-1)} \circ \cdots \circ g_\theta^{(1)} - g_\theta^{(k)} \circ g_\theta^{(k-1)} \circ \cdots \circ g_\theta^{(1)}\| \right) \tag{18}$$

$$= \sum_{k=1}^{L} \left( \prod_{l=k+1}^{L} \beta^{(l)} \|g_\theta^{(k)} \left( h_i^{(k-1)}, \bar{h}^{(k-1)} \right) - g_\theta^{(k)} \left( h_i^{(k-1)}, h^{(k-1)} \right)\| \right) \tag{19}$$

$$\le \sum_{k=1}^{L} \left( \prod_{l=k+1}^{L} \beta^{(l)} \| \sum_{\mathcal{N}(i)} \tilde{\hat{A}}_{i,} * \bar{h}^{(k-1)} - \sum_{\mathcal{N}(i)} \tilde{\hat{A}}_{i,} * h^{(k-1)}\| \right) \tag{20}$$

$$\le \sum_{k=1}^{L} \left( \prod_{l=k+1}^{L} \beta^{(l)} |\mathcal{N}(i)| * \|\tilde{\hat{A}}_{i,} * \bar{h}^{(k-1)} - \tilde{\hat{A}}_{i,} * h^{(k-1)}\| \right) \tag{21}$$

$$\le \sum_{k=1}^{L} \left( \prod_{l=k+1}^{L} \beta^{(l)} |\mathcal{N}(i)| * \|\tilde{\hat{A}}_{i,}\| * \|\bar{h}^{(k-1)} - h^{(k-1)}\| \right) \tag{22}$$

$\square$

