# OpenReview forum: "VISAGNN: Versatile Staleness-Aware Training for Efficient Large-Scale GNNs"
_ICLR.cc/2025/Conference — Submitted to ICLR 2025_

### Official Review · Reviewer_ZfEx · 2024-11-02

**Soundness:** 2
**Presentation:** 3
**Contribution:** 2
**Rating:** 3
**Confidence:** 4

**Summary:**

This paper is an intersting attempt to extend the historical-embedding-based method ReFresh (or FreshGNN, its official name in VLDB'24, which I will use in the following review).  It incorporates the staleness criteria in FreshGNN to the training procedure: the attention mechanism in message passing, loss function and the historical embedding itself.  The paper than shows some results on accuracy, speed and convergence speed for different methods and datasets.

**Strengths:**

In order to better understand this paper, I first went through FreshGNN carefully and find some improvements of FreshGNN in this paper:

1. The paper innovatively incorporated the staleness measurements into the training procedure, enabling dynamic employment of the historical embedding in the training.
2. The accuracy is improved compared to the baseline methods.

**Weaknesses:**

1. The techniques used by the authors are more from an empirical view. As a theoretical paper, I would like to see the paper be backed by more sound theoretical analysis.
2. In line 320 of page 6, the authors claimed "the model parameters tend to converge, resulting in smaller gradient values and fewer updates to the embeddings in later epochs", but I didn't see illustrations (experiments or figures) supporting this claim.  Similarly, "the influence of staleness diminishes as training progresses" is not supported by experiments.  (I am not denying these claims, but want to see evidence.)
3. In 3.2 (2) Centrality, it is not mentioned that which centrality measurement is used, and that the reasons or theoretical considerations of using and choosing this one better than not using or other ones.
4. This weakness is subtle. In FreshGNN, the gradient norm is only used for comparison within the same batch, so these gradients are calculated from the same weight parameter.  However, in VISAGNN, the gradient norms are used in the attention score calculation (eq. 5), here it is possible to have comparison/calculation of gradient norms calculated from different generations of the weight parameter.  Moreover, the grad norms are used in the "staleness-augmented embeddings" as part of the feature, but they also may come from different generations of weight parameter.

   Per the claim in weakness 2, which I do agree, the gradient values will become smaller as the training goes on.  Then is it valid to compare gradient norms across different generations of weight parameters?  Even if the historical embedding is away from the real embedding for the same distance, I would expect the one from later stage of the training to have a smaller magnitude in grad norm.  Please convince me why it is valid to compare the grad norms generated from different generations of weight parameters, and why it does not conflict with the claims mentioned in weakness 2.
5. In the experiment part, other large-scale datasets like MAG240M and IGB260M is absent.  I would like to see how VISAGNN performs on these datasets.

6. In the illustration of time comparison in Table 3, it is said "we exclude system-level optimizations from ReFresh to ensure a fair comparison".  I do not see how this is fair.  These are very important components in accelerating training speed.  GAS can be slower to SAGE if the SAGE is backed by DGL because of the system optimizations in DGL.  Sandbagging the baseline make the experimental much less convincing.  If the statistical proverty like convergence speed in terms of iterations were to be compared, the author can compare the epochs / iterations before converging, rather than doing the time comparison awkwardly.
7. Table 5 is confusing to me.  FreshGNN is designed for neighbor sampling specifically, why is it put in a senario of METIS clusters?  This use case seems to be out of the scope of FreshGNN.  Is there any adaption to FreshGNN done here?   Also, it is not clear which way of sampling that VISAGNN targets at.  Is it neighbor sampling like FreshGNN?  Or subgraph sampling like GAS, FM, LMC?

**Questions:**

1. The ogbn-prodcts accuracy for FreshGNN is 78.7 in Table 1, but it is 78.87 in the original FreshGNN paper.  Is it a typo?

2. Can you explain why VISAGNN take more memory than the baselines?  Also, the experiment setup is very vague.  How is the historical embedding cache constructed?  Is there a fixed size?  Or use the same eviction rule as FreshGNN?  Plus, FreshGNN uses all the available memory for feature caching, how is that calculated?

I would like to raise my score if all the weaknesses and questions are properly addressed.

---

### Official Review · Reviewer_hhrw · 2024-11-03

**Soundness:** 2
**Presentation:** 2
**Contribution:** 2
**Rating:** 3
**Confidence:** 4

**Summary:**

This paper introduces VISAGNN to address staleness in GNN training. VISAGNN proposes a dynamic staleness-aware attention mechanism, a staleness-aware regularization term, and staleness-augmented embeddings, allowing it to adapt to stale embeddings and improve performance on large datasets. Experimental results show its effectiveness compared to other GNN methods.

**Strengths:**

1. The framework shows improvements in accuracy and efficiency over existing GNN methods.

2. VISAGNN is designed to be adaptable and compatible with various existing GNN training frameworks.

**Weaknesses:**

1. The datasets used to validate VISAGNN, while relatively large, do not fully substantiate the scalability claims, especially when there exist much larger benchmarks, such as ogbn-papers100M. Without experiments on such datasets, it is challenging to conclude that VISAGNN can handle truly large-scale graph structures effectively.

2. The bound derived in Theorem 1 appears to be overly relaxed due to its formulation as a summation over all layers with products of per-layer values. This relaxation may limit the theorem’s practical utility in providing actionable insights for model design or optimization.

3.  Some recent, relevant work on handling staleness in GNN training is missing from the literature review. Notably, "SANCUS: Staleness-Aware Communication-Avoiding Full-Graph Decentralized Training in Large-Scale Graph Neural Networks" and “Staleness-Alleviated Distributed GNN Training via Online Dynamic-Embedding Prediction” directly addresses similar issues in distributed GNN training and should be discussed to contextualize VISAGNN better.

**Questions:**

Please refer to my points in the weaknesses section.

---

### Official Review · Reviewer_mjj9 · 2024-11-03

**Soundness:** 2
**Presentation:** 2
**Contribution:** 2
**Rating:** 5
**Confidence:** 4

**Summary:**

This paper proposes a VersatIle Staleness-Aware GNN, named VISAGNN to address the staleness issue of the historical embeddings. The key idea of VISAGNN is to embed staleness into the message-passing mechanism, loss function, and historical embeddings during training, making the model adaptively mitigate the negative effects of stale embeddings. Experiments demonstrate that VISAGNN outperforms existing historical embedding techniques in terms of efficiency and accuracy on large-scale benchmarks.

**Strengths:**

1. VISAGNN can scale GNN on ogbn-papers100M, which is a large-scale graph with more than 100 million nodes.
2. The proposed VISAGNN can be integrated with various historical baselines.
3. This paper is easy to follow.

**Weaknesses:**

1. The authors may want to demonstrate the effectiveness of VISAGNN on various GNN backbones, such as GCN, SAGE, and PNA.
2. The authors may want to report the standard deviation across replicates in Tables 1,2,3,4 and Figures 2,3.
3. "Accuracy (%)" in Figures 2,3 may be "Accuracy".

**Questions:**

1. Does the proposed method apply to homophily graphs?

---

### Official Review · Reviewer_GdvW · 2024-11-07

**Soundness:** 3
**Presentation:** 3
**Contribution:** 2
**Rating:** 3
**Confidence:** 5

**Summary:**

This paper is built based on historical embeddings based GNN training methods for large-scale graphs. One main limitation of these methods is the staleness of these historical embeddings. The authors propose a new method to overcome this limitation, including new message-passing model design (dynamic staleness attention),  loss function (staleness-aware loss), and node embeddings (staleness-augmented embeddings).

**Strengths:**

1. This paper is well-written and easy to follow.
2. The motivation is clear, and the proposed method improves existing methods from several aspects, including new design of GNN model, loss function, and node features.

**Weaknesses:**

1. The main weakness of this paper is the improvement over previous methods. The abstract mentions that the proposed method achieves “superior performance and efficiency on large-scale benchmarks, as well as significantly accelerated convergence”. However, from Table 1, the performance improvement is marginal.
2. About efficiency: why a little more memory and less time than GAS, as shown in Table 3.
3. Two related papers [1][2] are not mentioned and compared.
4. In addition to the staleness problem, I think these historical embeddings based methods have another main issue: the memory cost for the historical embeddings. This will be more challenging when applying to large-scale graphs, limiting their application to real-world applications.

[1] Staleness-Alleviated Distributed GNN Training via Online Dynamic-Embedding Prediction

[2] Staleness-based subgraph sampling for large-scale GNNs training

**Questions:**

see weaknesses

---

### Meta-Review · Area_Chair_jYyX · 2024-12-20

**Metareview:**

This paper proposes a VersatIle Staleness-Aware GNN, namely VISAGNN, to dynamically and adaptively incorporate staleness criteria into the large-scale GNN training process. While the idea of addressing staleness in GNN training is interesting, all reviewers tend to reject this paper due to several critical weaknesses, including the limited novelty, the insufficient experiments, and the marginal performance improvement over baselines.

First, the comparison with recent studies that address staleness in GNN training (e.g., [1,2,3]) is missing in this paper. Without these comparisons, it is difficult to evaluate the contribution of this paper. Second, the experiments on large-scale datasets such as ogbn-papers100M, MAG240M, and IGB260M are missing. These datasets are essential for demonstrating the scalability and practical applicability of the proposed method in the real world. Third, as highlighted by the reviewers, the performance improvement shown in Table 1 is marginal, which limits the contribution of this paper.

Furthermore, the authors did not submit a rebuttal or provide any additional clarifications, leaving these concerns unaddressed. Given these significant weaknesses, I cannot recommend the acceptance of this paper in its current form.


[1] Staleness-Alleviated Distributed GNN Training via Online Dynamic-Embedding Prediction.

[2] Staleness-based subgraph sampling for large-scale GNNs training.

[3] SANCUS: Staleness-Aware Communication-Avoiding Full-Graph Decentralized Training in Large-Scale Graph Neural Networks.

**Additional Comments On Reviewer Discussion:**

Reviewers hhrw, mjj9, ZfEx, GdvW rated this paper as 3: reject, 5: borderline reject, 3: reject, and 3: reject, respectively.


All the reviewers tend to reject this paper and have pointed out some important weaknesses, including the limited novelty, the insufficient experiments, and the marginal performance improvement over baselines. First, the comparison with recent studies that address staleness in GNN training is missing in this paper. Without these comparisons, it is difficult to evaluate the contribution of the proposed method. Second, the experiments on large-scale datasets such as ogbn-papers100M, MAG240M, and IGB260M are missing. These datasets are essential for demonstrating the scalability and practical applicability of the proposed method in the real world. Third, as highlighted by the reviewers, the performance improvement shown in Table 1 is marginal, which limits the contribution of the method. However, the authors did not respond to the reviews in the author-reviewer discussion period.

---

### Decision · Program_Chairs · 2025-01-22

Reject